# Evaluation of the Impact of Serum Vitamin D Levels on the Scoring Atopic Dermatitis Index in Pediatric Atopic Dermatitis

**DOI:** 10.3390/children10091522

**Published:** 2023-09-07

**Authors:** Fatih Çiçek, Mehmet Tolga Köle

**Affiliations:** 1Department of Pediatric Allergy and Immunology, Kartal Dr. Lütfi Kırdar City Hospital, University of Health Sciences, Istanbul 34870, Turkey; 2Department of Pediatrics, Kartal Dr. Lütfi Kırdar City Hospital, University of Health Sciences, Istanbul 34870, Turkey; mehmettolga.kole@saglik.gov.tr

**Keywords:** SCORAD, atopy, atopic dermatitis, eosinophilia, serum vitamin D

## Abstract

Atopic dermatitis (AD) is a chronic and recurrent inflammatory skin condition characterized by itching, eczematous plaques, and dry skin. Despite ongoing research, its exact cause remains elusive. In this study, we aimed to explore the factors that influence the severity of AD in children and assess the relationship between serum vitamin D levels and the disease’s severity. We enrolled 96 AD patients in our investigation, evaluated their clinical condition using the Scoring Atopic Dermatitis (SCORAD) index, and compared them to a group of 90 healthy controls. Our analysis revealed that serum vitamin D levels and eosinophil counts significantly impacted the SCORAD index (*p* < 0.001). According to standardized regression coefficients, for each incremental unit in serum vitamin D levels, the SCORAD index exhibited a decrease of 0.449 units. Similarly, a one-unit increase in eosinophil count resulted in a 0.009 unit increase in the SCORAD index. It is worth noting that the influence of serum vitamin D levels on disease severity surpasses that of eosinophil counts and atopic conditions. In our patient cohort, we uncovered a negative correlation (r = −0.419, *p* < 0.001) between serum vitamin D levels and the SCORAD index. Our findings suggest that low serum vitamin D levels may have a more substantial impact on AD severity than atopic conditions and eosinophilia. Furthermore, we observed a negative association between the severity of AD and serum 25(OH)D3 levels.

## 1. Introduction

Atopic dermatitis (AD) is a chronic and recurrent inflammatory skin disorder characterized by itching, dry skin, and eczematous plaques [1,2]. The pathogenesis of AD is not completely known but is associated with environmental, genetic, biochemical, and immunological factors [3,4]. The incidence is higher in metropolitan families and provinces with high socioeconomic standing, and the prevalence is roughly 15% to 30% in children and 2% to 10% in adults [5,6]. The disease manifests itself predominantly in early childhood; 45% of patients develop symptoms before six months, and 60% of patients before their first birthday [7].

The disorder severity can be assessed with the Scoring Atopic Dermatitis (SCORAD) index [8]. The SCORAD index consists of three criteria: (A) extent (percentage of skin surface affected according to the rule of nine); (B) intensity (incorporation of erythema, edema/papules, the outcome of itching, oozing/crust appearance, lichenification, and dryness, six parameters evaluated on a scale of 0–3; 0: absent, 1: mild, 2: moderate, 3: severe); and (C) subjective symptoms (for the last three days, the severity of pruritus, the impact of sleep, and the overall condition of the skin on daily life are questioned, and the responses are evaluated on a scale of 1 to 10) [9]. The formula A/5 + 7B/2 + C is applied to all the obtained data, and the SCORAD score is calculated. If the index score falls between 0 and 24.9, it is categorized as mild; between 25 and 50 is moderate; and above 50 is severe atopic dermatitis [9].

In addition to the well-described function in calcium homeostasis and immunity, vitamin D has miscellaneous skin functions, such as keratinocyte proliferation, skin barrier function, and apoptosis [10]. In addition, vitamin D may affect autoimmune diseases, infections, malignancies, cardiovascular diseases, neuropsychiatric disorders, and allergic disease courses, including AD [11,12]. It is believed that the anti-inflammatory effect of vitamin D plays a role in its therapeutic efficacy in modulating disease progression [12] In recent years, the significance of vitamin D’s anti-inflammatory effect has been highlighted multiple times [12,13]. In a study conducted by Karampinis et al. low serum vitamin D levels have been shown to lead to higher rates of exacerbations in patients with psoriasis [14]. Reinholz et al. emphasized that lower serum 25-hydroxyvitamin D3 (25(OH)D3) conditions are associated with higher IgE sensitivity, food allergy, and asthma [15]. Another study reported that the SCORAD index score has a negative correlation with vitamin D [16]. However, other studies have emphasized that they have not found a relationship between AD and vitamin D [17,18]. The effect of vitamin D levels on the course of AD is controversial; therefore, we aimed to evaluate the factors affecting the severity of AD in children and to evaluate the relationship between serum vitamin D levels and the severity of AD.

## 2. Materials and Methods

Ninety-six patients diagnosed with AD and 90 healthy controls without chronic systemic or allergic diseases were enrolled in our case–control study, which was conducted from September 2022 to November 2022.

We collected clinical evaluation, SCORAD index data, and laboratory analysis results, including complete blood count, food, and aero allergen serum-specific immunoglobulin E (IgE), serum total IgE level, and serum 25(OH)D3 concentration, from patient records retrospectively. The patients were diagnosed with AD according to the Hanifin and Rajka criteria by a pediatric allergist. Atopic dermatitis severity was assessed using the SCORAD index. Based on this index, AD was categorized into mild AD (0–24.9), moderate AD (25–50), and severe AD (>50) [9].

We excluded the patients who used systemic glucocorticoid therapy the previous month and used vitamin D supplements the prior six months from the study. Also, we excluded patients who received local steroid treatment from the study as the current treatment might affect the eczema harshness and the SCORAD index. In addition, we excluded patients receiving multi-vitamin supplements and drugs that have the potential to affect serum 25(OH)D3 levels (like antacids or anticonvulsants or systemic steroid therapy), patients suffering from a systemic disorder or with clinical signs consistent with rickets-like limb deformities, obese individuals, patients with upper respiratory tract illness, chronic lung disease (asthma or others), an impaired immune system, neurological or metabolic diseases. The participants’ serum vitamin D levels were evaluated in the autumn. Participants living in the same region had similar levels of sunlight exposure. Sunlight is an important source of vitamin D [19]. The dietary characteristics of the patients were not evaluated. Those with malnutrition were excluded from the study.

### 2.1. Skin Prick Tests

For forearm skin prick tests, we evaluated patients who had not received antihistamines for seven days, applying serum histamine (10 mg/mL) and 0.9% saline as positive and negative references, respectively. We gauged responses at the fifteenth minute. We defined the positive reaction as ≥3 mm above the negative control [20].

We performed the prick tests with the same antigens (egg yolk, egg white, milk, peanut, hazelnut, walnut, wheat, fish, soy, *Dermatophagoides farinea*, *Dermatophagoides pteronyssinus*, *Alternaria alternata*, dog and cat dander, cockroaches–mix, and grass–mix, grain–mix, tree–mix, and weed–mix pollens) (Lofarma S.p.A, Milan, Italy) by using Stallerpoint (Allertech, Doral, FL, USA).

### 2.2. Total Eosinophil Count, Specific IgE in Serum, Total IgE, and Atopy

We obtained peripheral blood eosinophil counts from hemogram parameters and measured total IgE from serum (by nephelometric method with a machine from Siemens Healthcare Diagnostics Products, Marburg, Germany) and assayed causal or suspected food (milk, egg white, egg yolk, peanut, hazelnut, walnut, wheat, fish, soy) and aero (*Dermatophagoides farinea*, *Dermatophagoides pteronyssinus*, *Alternaria, alternata*, *Aspergillus*, cockroaches mix, cat and dog dander, and grass-mix and tree-mix pollens) allergen-specific IgE levels with the Immuno-CAP system (UniCAP; Uppsala, Sweden) (≥0.35 kIU/L was considered to be positive). The patient’s atopy condition was described by positivity to at least one allergen by skin prick test (SPT; wheal ≥3 mm larger than negative control) or specific IgE testing (≥0.35 kIU/L) [20]. Individuals with no positive results were classified as non-atopic (non-sensitized).

### 2.3. 25-Hydroxyvitamin D3 Levels

We took peripheral venous blood samples at 6–10 h fasting and measured serum 25-(OH)D3 levels used the liquid chromatography-tandem mass spectrometry procedure (Waters Quattro Premier XE TM, Waters Corp, Milford, MA, USA). Although the most active form in the blood is 1–25(OH) vitamin D, its half-life is quite short, about 3–4 h [21]. In contrast, the half-life of 25-(OH)D3 is on average about 3 weeks, and this form accumulates mainly in adipose tissue [21]. Digested and synthesized by the skin, vitamin D is rapidly converted to 25-(OH)D3, but only a fraction of the 25-(OH)D3 in serum is converted to the active metabolite 1,25-(OH) vitamin D. Serum levels of 1,25-(OH) vitamin D are very low or there is no correlation with vitamin D stores [21,22]. The concentration of 25(OH) vitamin D3 in serum is higher than 1–25(OH) vitamin D3 [22]. For these reasons, we used the 25(OH)D3 form of vitamin D to assess serum levels in our study. Serum 25-hydroxyvitamin D3 levels ranging from 10.1 to 24 ng/mL were considered mild to moderate vitamin D deficiency, levels between 24.1 and 80 ng/mL were considered optimal, and levels above 80 ng/mL were considered toxic [23]. We evaluated serum 25-hydroxyvitamin D3 levels between September and November 2022.

### 2.4. Statistical Analysis

We conducted the data analysis employing IBM SPSS Statistics Standard Concurrent User V 26 and MedCalc^®^ Statistical Software version 19.6, both widely recognized statistical package applications. Descriptive statistics are reported as the count of instances (n), proportions (%), mean ± standard deviation, and median (with interquartile range). To assess data normality, we employed the Shapiro–Wilk test and verified the homogeneity of variances using the Levene test. When dealing with non-normally distributed continuous variables, we resorted to the Mann–Whitney U test for between-group comparisons. For assessing relationships between deficiency indices and continuous variables, we computed the Spearman correlation coefficient. When examining the associations between categorical variables, we employed both the Pearson chi-square test and Fisher’s exact test. To delve deeper into our analysis, we employed multiple linear regression analysis. This allowed us to investigate the impact of serum vitamin D deficiency, eosinophilia, and atopy on the SCORAD index. We rigorously assessed the suitability of the linear regression model, taking into account various diagnostic checks. These checks included the Shapiro–Wilk test to evaluate residual normality, Durbin–Watson statistics to assess autocorrelation, variance inflation factor statistics, and the tolerance test to gauge linearity assumptions. We affirm that all necessary assumptions were met for the two regression models we constructed. A significance level of *p* < 0.05 was deemed statistically significant. Formun Üstü

### 2.5. Ethics

We obtained the study approval from the ethics committee of Kartal Dr. Lütfi Kırdar Training and Research Hospital (Date: 26 October 2022, Number: 2022/514/236/12) and conducted the study in accordance with the Helsinki criteria.

## 3. Results

We enrolled a total of 186 participants in our study, comprising 90 individuals in the control group and 96 patients, all of whom were assessed during the autumn season. There were no notable disparities in terms of gender (*p* = 0.999) and age (*p* = 0.855) distribution between the two groups. However, the patient group exhibited significantly higher eosinophil and total IgE levels, while their vitamin D levels were notably lower (*p* = 0.018 and *p* = 0.001, respectively) (Table 1).

In 56 of the patients (equivalent to 58.3%), there was an evident deficiency in vitamin D levels. Despite the data presented in Table 2, which indicate that females had a slightly higher median SCORAD index compared to males, this disparity did not reach statistical significance (*p* = 0.072). However, a notable discrepancy was observed in the SCORAD index concerning vitamin D status (*p* < 0.001). Specifically, the group afflicted by severe vitamin D deficiency exhibited a markedly elevated SCORAD index in contrast to the mild to moderate and optimal groups (*p* < 0.001). In contrast, there was no statistically noteworthy contrast between the mild to moderate and optimal groups. Furthermore, it is worth noting that the SCORAD scores in the sensitive cohort were significantly elevated compared to those in the non-sensitive group (*p* = 0.002). Similarly, within the subset of individuals with food allergies, the SCORAD index demonstrated a substantial increase in comparison to the group void of food allergies (*p* = 0.002). Nonetheless, among individuals with specific food allergies (such as milk, egg white, and egg yolk), the SCORAD index exhibited a statistically significant elevation in comparison to those lacking such allergies (respectively, *p* = 0.009, *p* = 0.002, *p* = 0.037) (refer to Table 2 for details). Nevertheless, we did not uncover any noteworthy distinctions in the SCORAD index among those with allergies to other food items (such as peanuts, hazelnuts, walnuts) (*p* > 0.05) or aeroallergens (such as mites, pollen, cat, dog epithelium) (*p* = 0.422, *p* = 0.204, *p* = 0.512, respectively).

We observed a noteworthy (*p* = 0.001) positive association between the overall eosinophil count and the SCORAD index (r = 0.336). However, the correlation coefficients between age, gender, total IgE, and the SCORAD index did not reach statistical significance (*p* = 0.064, *p* = 0.194, respectively) (Table 3).

The effect of serum vitamin D levels, eosinophil count, and atopy on disease severity was evaluated by multiple linear regression analysis (Table 4). Gender and age were not considered in the analysis as they were confounding factors. We determined the final factors affecting disease severity by the stepwise method. In the regression analysis, atopy was excluded from the model, and as a result, serum vitamin D levels and eosinophil counts were found to be effective on the SCORAD index (*p* < 0.001). According to Table 4, as serum vitamin D level increases by one unit, the SCORAD index decreases by 0.449 units, and as eosinophil value increases by one unit, the SCORAD index increases by 0.009 units. According to standardized regression coefficients (absolute values), the effect of serum vitamin D levels on disease severity is greater than that of eosinophil counts.

As shown in Table 5, AD severity was identified as mild in 33 (34.3%), moderate in 52 (54.2%), and severe in 11 (11.5%) cases. Furthermore, we observed a significant disparity in serum vitamin D levels among the groups with different disease severities (*p* < 0.001). Serum vitamin D levels were statistically lower in the severe AD group compared to the mild and moderate groups. However, the difference in serum vitamin D levels between the mild and moderate groups did not reach statistical significance (Table 5).

In all patients, we found a statistically significant (*p* < 0.001) moderately negative correlation (r = −0.419) between serum vitamin D levels and the SCORAD index (Figure 1). Moreover, within the atopic group, a statistically significant (*p* = 0.001) moderate (r = −0.460) negative association was observed between vitamin D and the SCORAD index. Additionally, in patients without atopy, there was a statistically significant (*p* = 0.031) weak negative correlation (r = −0.305) between vitamin D and the SCORAD index.

## 4. Discussion

Easily accessible and practical laboratory tests can serve as valuable tools for assessing the severity of AD [24]. In this context, we investigated the relationship between serum vitamin D levels and the severity of AD, as well as the impact of other factors. Our study’s findings underscore an inverse correlation between serum vitamin D levels and the severity of AD in children. This highlights the substantial influence of vitamin D deficiency on AD severity, surpassing the effects of atopy and eosinophilia. The well-established roles of vitamin D in immunomodulation and calcium homeostasis come into play here [10,25]. Vitamin D plays a pivotal role in modulating the production of cytokines, particularly those involved in the Th1/Th2 balance [26]. Th2 cytokines, such as interleukin-4 (IL-4), IL-5, and IL-13, are relevant to the pathogenesis of AD and may also have a regulatory impact on vitamin D production [26]. Also, vitamin D has antimicrobial properties, and its deficiency has an association with increased susceptibility to infections [25]. Skin colonization by Staphylococcus aureus has an association with the development and exacerbation of AD, and vitamin D may reduce bacterial colonization on the skin [25]. Recent studies have underlined vitamin D’s potential role in AD, and its supplementation can support AD treatment.

Most vitamin D is obtained through exposure of the skin to sunlight. Sunlight, especially ultraviolet B (UVB) radiation, has a significant immunomodulatory effect on the skin. Exposure to UVB radiation stimulates the production of vitamin D in the skin, which plays a critical role in immune system regulation [27]. It is observed that at higher geographic latitudes, where there is less sunlight exposure and reduced vitamin D production, there is a higher prevalence of AD [28]. Vitamin D can influence not only calcium regulation but also immunomodulation. Specifically, vitamin D can affect local calcium balance and regulate gene transcription by targeting vitamin D receptors that regulate immune system responses [29]. Additionally, vitamin D has been shown to inhibit the release of markers such as IL-6 and tumor necrosis factor-alpha (TNF-α), which promote inflammation and worsen AD symptoms [30]. Moreover, vitamin D is highlighted for its ability to enhance the production of antimicrobial peptides in the skin, strengthening the skin’s natural defense and thereby reducing the risk of infection or exacerbation [31]. Furthermore, it is believed that vitamin D may contribute to the maintenance of skin integrity by supporting the production of critical proteins like filaggrin and involucrin, which are essential for skin barrier function [10].

The relationship between vitamin D levels and the severity of AD remains a contentious topic in the existing literature. In our study, the mean serum 25(OH)D3 levels were notably lower in children with AD when compared to the control group. This observation aligns with a study by Wang et al., which involved 498 pediatric AD patients and 328 individuals in a control group; they also reported statistically lower average serum levels of 25(OH)D3 in the AD group [5]. Our findings are consistent with similar investigations that compared children with AD to healthy control groups, consistently finding lower mean serum 25(OH)D3 levels in the patient groups [32,33]. In contrast to our results, two separate studies among children did not reveal a statistical difference between the AD group and the control group in terms of serum 25(OH)D3 levels [2,34]. It is worth noting that in one of these studies, there were disparities in the evaluation seasons of the participants, especially considering that the average vitamin D values were significantly lower than those observed in our study [2]. However, a meta-analysis encompassing 11 studies demonstrated a significant difference between the patient and control groups [35]. This implies that the risk of vitamin D deficiency in pediatric AD might be considerably higher, consistent with the findings of a meta-analysis conducted by Kim and colleagues, which also indicated an elevated risk of vitamin D deficiency in pediatric AD [36].

Our findings suggest that individuals with severe AD exhibit lower levels of vitamin D compared to those with mild and moderate AD. Furthermore, we have identified a strong negative correlation between serum vitamin D levels and the severity of AD. These results are consistent with prior research, which consistently reported an inverse relationship between serum vitamin D levels and the severity of AD [5,16,34,37]. In a recent study conducted by Bulut et al., a negative correlation was observed between serum 25(OH)D3 levels and the severity of AD in 120 pediatric patients, which is in agreement with our findings [4]. Additionally, in line with the outcomes of our study, serum 25(OH)D3 levels were statistically lower in patients with severe AD compared to other groups [4]. Similar studies in the existing literature have reported a negative association between disease severity and serum 25(OH)D3 levels [2,5,16,34,37]. However, it is worth noting that some studies, in contrast to our findings, did not establish a correlation between disease severity and serum 25(OH)D3 levels [17,18,38]. Raj et al. also found a negative correlation between serum vitamin D levels and disease severity. They emphasized that vitamin D supplementation had the most beneficial impact in severe atopic dermatitis cases and the least in mild cases [34]. Considering these findings, assessing serum vitamin D levels could be advisable for severe AD patients or cases where disease severity does not improve as expected despite appropriate treatment. To gain further insights, large-scale, multicenter, prospective studies are necessary to clarify this relationship. Another aspect worthy of investigation is whether vitamin D deficiency contributes to increased skin lesions due to its detrimental effect on barrier function, or if the deficiency arises as a consequence of reduced vitamin production in diseased skin.

We observed a significant increase in the SCORAD index in the group without sensitivities compared to the non-sensitive group. Although the SCORAD index was notably higher in the food allergy group, we did not find a significant impact of inhaled allergen sensitivity on the SCORAD index. Similarly, Lee et al. reported elevated SCORAD indices in patients with food allergies [38]. In a study conducted within our nation, Akan et al. uncovered that serum 25(OH)D3 levels displayed an adverse connection with the seriousness of AD within the cohort characterized by allergic sensitivities. Conversely, no discernible correlation emerged within the group devoid of sensitivities [39]. In our investigation, we detected a statistically significant albeit slight inverse relationship between serum 25(OH)D3 levels and the SCORAD index among individuals lacking sensitivities. Epidemiological investigations have spotlighted the role of vitamin D in the development of sensitivities [40]. Vitamin D impedes the proliferation of T cells and contributes to the conversion of CD4+ T cells into regulatory T cells, which are acknowledged for their capacity to suppress pro-allergic mechanisms [41]. Consequently, we emphasize the importance of vigilantly monitoring serum 25(OH)D3 levels, particularly in AD patients with sensitivities.

In our study, we found that serum vitamin D deficiency had a greater impact on the severity of AD than atopy. Although the existing literature on this subject is limited, Baek et al. found a similar correlation between the SCORAD index and serum vitamin D levels when compared to atopy, which aligns with our findings [42]. In this study, Baek et al. found that low 25(OH)D3 levels increased food allergen sensitivity risk and were related to the severity of AD [42]. Eosinophils are closely linked with the effector component of immune response mediated by type 2 T helper cells, which plays a pivotal role in allergic reactions [43]. Specifically, 1,25(OH) vitamin D has been shown to extend the lifespan of eosinophils and enhance the expression of CXCR4, a receptor involved in chemotaxis and immune regulation [44]. Additionally, it has been observed that vitamin D can reduce eosinophil necrosis and limit the release of cytolytic peroxidase [45]. Nevertheless, the ongoing debate revolves around the association between serum vitamin D levels and the eosinophil count in the bloodstream. While a few studies have established a link between lower vitamin D levels in the blood and elevated eosinophilia, the majority of research has not reported a significant correlation [46,47,48]. In our investigation, we uncovered a statistically significant positive connection between the SCORAD index and the overall eosinophil count, whereas no notable correlation emerged regarding factors like age, gender, total IgE, and the SCORAD index. Correspondingly, Cheon’s study and the findings of Baek and colleagues corroborated our results by identifying affirmative correlations between the SCORAD index and the total eosinophil count [33,42]. Furthermore, in line with our findings, studies conducted by Su and team, as well as Baek and others, failed to establish a connection between total IgE levels and the SCORAD index [2,42]. When evaluating the impact of serum vitamin D levels and eosinophils on disease severity, we observed that serum vitamin D levels had a more pronounced effect on disease severity in comparison to eosinophilia. For every unit increase in serum vitamin D levels, we observed a reduction of 0.449 units in the SCORAD index. Conversely, for each unit increase in eosinophil count, we noted a mere 0.009 unit increase in the SCORAD index. To clarify this relationship, we are of the opinion that comprehensive, multi-center studies are essential.

### Limitations and Strengths

One of the primary limitations of our study is the inability to assess factors that could influence the severity of atopic dermatitis, such as environmental pollutants or psychosomatic factors. The limitation of being a single center and having a small sample size can be considered as other limitations of our study. Comprehensive assessment of the impact of vitamin D deficiency, eosinophil count, and atopy on AD is the strength of the study. We established that serum 25(OH)D3 deficiency exhibited a stronger predictive value for the SCORAD index compared to eosinophilia and atopy in a regression model. This is another strength of the study.

## 5. Conclusions

Based on our analysis of the findings, it appears that reduced serum vitamin D levels could exert a more significant influence on the severity of AD when contrasted with atopy and eosinophilia. We identified an inverse relationship between AD severity and serum 25(OH)D3 levels. In essence, this implies that individuals with lower serum 25(OH)D3 levels may encounter more pronounced symptoms of the disease.

## Figures and Tables

**Figure 1 children-10-01522-f001:**
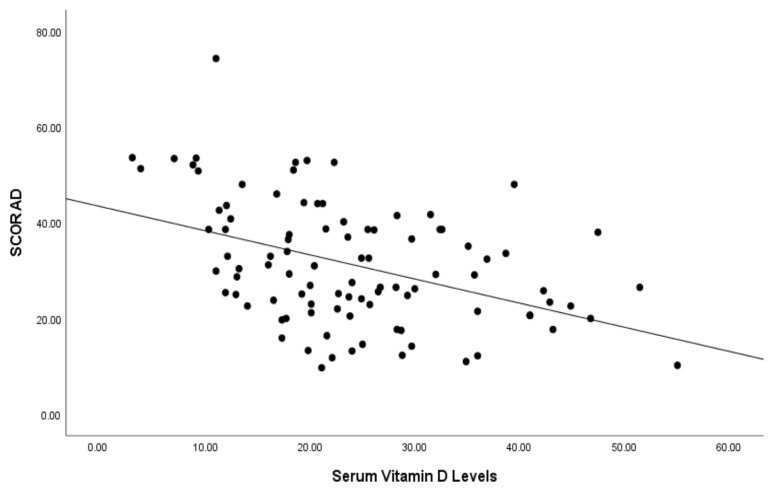
Correlation of serum vitamin D levels (ng/mL) and SCORAD index of patients.

**Table 1 children-10-01522-t001:** Comparison of variables by group.

	Group		
	Control *n* = 90	Patient *n* = 96	*p* Value
Gender, *n* (%)			
Female	43 (47.8%)	45 (46.9%)	0.999 ^&^
Male	47 (52.2%)	51 (53.1%)	
Age (years)	4 (2–8)	4 (3–6)	0.855
Total eosinophil count (10^3^/mm^3^)	287.4 ± 45.2	444.7 ± 43.8	0.018 ^†^
Log total IgE	1.71 ± 0.080	1.99 ± 0.07	0.018 ^†^
25-hydroxyvitamin D3 (ng/mL)	29.71 ± 1.19	23.68 ± 1.15	0.001 ^†^

*n*: Number of patients; %: Column percent, numerical data are given as median (interquartile range) or mean ± standard error; ^&^: Pearson chi-square; Mann–Whitney U test; ^†^: One way analysis of covariance adjusting for age; LogIgE: Logarithm-transformed plasma total IgE level.

**Table 2 children-10-01522-t002:** Comparison of disease severity with vitamin D levels and atopy in the patient group (*n* = 96).

		SCORAD Index	
	*n* (%)	*M* (*IQR*)	*p* Value
Gender			
Girls	45 (46.9)	33.0 (18.1)	0.072 ^†^
Boys	51 (53.1)	26.5 (17.3)	
Vitamin D status			
Severe (≤10 ng/mL)	6 (6.2)	52.7 (2.3) ^a^	
Mild to moderate (10.1–24 ng/mL)	50 (52.1)	30.7 (18.3) ^b^	<0.001 ^‡^
Optimal (24.1–80 ng/mL)	40 (41.7)	26.0 (14.5) ^b^	
Sensitization			
None	50 (52.1)	25.6 (14.3)	0.002 ^†^
Positive	46 (47.9)	36.5 (18.8)	
Food allergy			
None	81 (84.4)	26.9 (17.9)	0.002 ^†^
Positive	15 (15.6)	40.8 (22.6)	
Aeroallergen sensitivity			
None	62 (64.6)	29.1 (17.5)	0.285 ^†^
Positive	34 (35.4)	31.7 (17.7)	

*M*: Median, *IQR*: Interquartile range, ^†^: Mann–Whitney U test, ^‡^: Kruskal–Wallis test, a and b superscripts indicate differences between D vitamin deficiency groups. There were no statistically significant differences between groups with the same superscripts.

**Table 3 children-10-01522-t003:** Correlations between SCORAD index and age, gender, total eosinophil count, and log total IgE.

	SCORAD İndex	
	*rho*	*p*
Age	−0.190	0.064
Gender	−0.178	0.098
Total eosinophil count	0.336	0.001
Log total IgE	0.134	0.194

*rho*: Spearman correlation coefficient.

**Table 4 children-10-01522-t004:** Evaluation of factors influencing the SCORAD index using multiple linear regression analysis.

	Regression Coefficients *						
	*β*	*se*	*Zβ*	*t*	*p*	95.0% Confidence Interval for *β*	
						*Lower Bound*	*Upper Bound*
Constant	39.198	3.665		10.694	<0.001	31.918	46.479
Serum vitamin D levels	−0.449	0.103	−0.383	−4.360	<0.001	−0.653	−0.244
Eosinophil Count	0.009	0.002	0.361	4.146	<0.001	0.005	0.013

Model summary: F = 11.301; *p* < 0.001; R^2^ = 0.332; Adj R^2^ = 0.303. Variables considered in the model: Serum vitamin D levels, atopy, eosinophil count; * Adjusted for gender, age; *β*: Regression coefficient; *Zβ*: Standardized regression coefficient; *se*: Standard error of *β*.

**Table 5 children-10-01522-t005:** Comparison of vitamin D levels regarding disease severity.

		25(OH)D, (ng/mL)	
	*n* (%)	*M* (*IQR*)	*p* Value
AD severity (SCORAD)			<0.001 ^‡^
Mild (≤25)	33 (34.3)	25.0 (15.4) ^a^	
Moderate (25–50)	52 (54.2)	22.1 (13.8) ^a^	
Severe (>50)	11 (11.5)	9.3 (11.6) ^b^	

*M*: Median; *IQR*: Interquartile range; ^‡^: Kruskal–Wallis test, a and b superscripts indicate differences between SCORAD groups. There were no statistically significant differences between groups with the same superscripts.

## Data Availability

Not applicable.

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
