# Peer review of "Evaluation of the Impact of Serum Vitamin D Levels on the Scoring Atopic Dermatitis Index in Pediatric Atopic Dermatitis"

_children, 2023, doi:10.3390/children10091522_

Round 1

Reviewer 1 Report

I read with great interest the manuscript proving a relation betweem serum Vitamin D Levels and  SCORAD Index in Pediatric Atopic Dermatitis. By comparing children with and without atopic dermatitis the authors concluded that there was a negative correlation  between serum vitamin D levels and SCORAD index in all patients while the low serum vitamin D levels may have a greater impact on the severity of the disease compared to  atopy and eosinophilia. It is a very interesting proposal for publications. However there are some flaws that need to be addressed.

1) there are many references for the term " atopy" please give a terminology

2)in the paragraph of introduction( lines 35-39) introducing the SCORAD please give an example of categorisation

3) in the paragraph of introduction( lines 40-51) referring to Vitamin D antinflammatory actions please add that Vitamin D can be protective against inflammatory derrmatosis flare ups as in the case of psoriasis. Use the following reference:

Karampinis E, Goudouras G, Ntavari N, Bogdanos DP, Roussaki-Schulze AV, Zafiriou E. Serum vitamin D levels can be predictive of psoriasis flares up after COVID-19 vaccination: a retrospective case control study. Front Med (Lausanne). 2023;10:1203426. Published 2023 May 25. doi:10.3389/fmed.2023.1203426

4) in the methods section:

please add why you used 25(OH) d3 to assess vitamin D status of the patients and not another metabolite

also refer to the system of vitamin D categorisation you used

did you monitored diet ( as diet is an additional source of vitmain D) if not refer it and add that sunlight is the main source

the citation you used (Meral, G.; Guven, A.; Uslu, A.; Can, G.; Yozgatli, AU.; Yaprak, P.; Akcay, F.. The prevalence of vitamin D deficiency in chil- 324 dren, adolescents and adults in a sample of Turkish population. Studies on Ethno Medicine 2016,10,249-25) refers to vitamin D defieciency prevelance in Turkish children.. not a vitamin D categorisation system. Using the above mentioned citation you can compare your findings of vitamin D with that found in the respective study

145 line : corrext the word SCORAD

correct the Figure 1: the text of the horizontal line 

maybe in the discussion section you should also write briefly about the cutaneous immunomodulatory effect of sunlight on skin as vitamin D main source is sunlight  

also in the discussion section you should add if vitamin D affects eosiniphils count: 

Souto Filho JTD, de Andrade AS, Ribeiro FM, Alves PAS, Simonini VRF. Impact of vitamin D deficiency on increased blood eosinophil counts. Hematol Oncol Stem Cell Ther. 2018;11(1):25-29. doi:10.1016/j.hemonc.2017.06.003

Author Response

Dear Reviewer,

Thank you for your interest in our manuscript entitled “Evaluation of the Impact of Serum Vitamin D Levels on the SCORAD Index in Pediatric Atopic Dermatitis”.  Our responses are outlined below. Our corrections were also made in the revised manuscript. Corrections were indicated as yellow.

Point 1: there are many references for the term " atopy" please give a terminology

Response 1: : Thank you for your valuable suggestions and comments.  ‘Atopy’ and ‘non-atopy’ terminologies are detailed and indicated as yellow.

‘The patient’s atopy condition was described by positivity to at least one allergen by skin prick test (SPT; wheal ≥3 mm larger than negative control) or specific IgE testing (≥0.35 kIU/l) [20]. Individuals with no positive results were classified as non-atopic (non-sensitized).’

Point 2: in the paragraph of introduction( lines 35-39) introducing the SCORAD please give an example of categorisation

Response 2: The categorization of the SCORAD index is outlined in the Materials and Methods section, (line 62). In addition, the part you recommended in the introduction has been added and indicated as yellow. ‘The SCORAD index consists of three criteria: extent (A) (percentage of skin surface affected according to the rule of nine), intensity (B) (incorporation of erythema, edema/papules, the outcome of itching, oozing/crust appearance, lichenification, and dryness, 6 parameters evaluated on a scale of 0-3; 0: absent, 1: mild, 2: moderate, 3: severe), and subjective symptoms (C) (For the last three days, the severity of pruritus, the impact of sleep, and the overall condition of the skin on daily life are questioned. The responses are evaluated on a scale of 1 to 10) [9]. The formula A/5 + 7B/2 + C is applied to all the obtained data, and the SCORAD score is calculated. If the index score falls between 0-24.9, it is categorized as mild; between 25-50 as moderate; and above 50 as severe atopic dermatitis [9].’

Point 3: in the paragraph of introduction( lines 40-51) referring to Vitamin D antinflammatory actions please add that Vitamin D can be protective against inflammatory derrmatosis flare ups as in the case of psoriasis.

Response 3: Thank you for your valuable suggestion. We emphasized the anti-inflammatory activity of vitamin D and indicated as yellow in introduction part. ‘It is believed that the anti-inflammatory effect of vitamin D plays a role in its therapeutic efficacy in modulating disease progression(34). In recent years, the significance of vitamin D's anti-inflammatory effect has been highlighted multiple times [12,13]. In a study conducted by Karampinis et al., low serum vitamin D levels have been shown to lead to higher rates of exacerbations in patients with psoriasis [14].’

Point 4: please add why you used 25(OH) d3 to assess vitamin D status of the patients and not another metabolite

Response 4: Thank you for your suggestion, we mentioned why we use the 25(OH)D3 form in the material method section indicated as yellow.

‘We measured serum 25- (OH) D3 levels with Milford, USA). Although the most active form in the blood is 1-25 (OH) vitamin D, its half-life is quite short, about 3-4 hours [21]. In contrast, the half-life of vitamin 25-(OH) D3 is on average about 3 weeks, and this form accumulates mainly in adipose tissue [21]. Digested and synthesized by the skin, vitamin D is rapidly converted to 25-(OH) D3, but only a fraction of the 25- (OH) D3 in serum is converted to the active metabolite 1,25- (OH) vitamin D. Serum levels of 1,25- (OH) vitamin D are very low or there is no correlation with vitamin D stores [21,22]. The concentration of vitamin 25(OH) D3 in serum is higher than vitamin 1-25 (OH) D [22]. For these reasons, we used the 25 (OH) D3 form of vitamin D to assess serum levels in our study.’

Point 5: also refer to the system of vitamin D categorisation you used. the citation you used (Meral, G.; Guven, A.; Uslu, A.; Can, G.; Yozgatli, AU.; Yaprak, P.; Akcay, F.. The prevalence of vitamin D deficiency in chil- 324 dren, adolescents and adults in a sample of Turkish population. Studies on Ethno Medicine 2016,10,249-25) refers to vitamin D defieciency prevelance in Turkish children.. not a vitamin D categorisation system.

Response 5: : Thank you for your attention. We have changed and added  a reference for the vitamin D deficiency categorization system.

Point 6: did you monitored diet ( as diet is an additional source of vitmain D) if not refer it and add that sunlight is the main source

Response 6: Thank you for your valuable comment. We excluded the patients who used vitamin D supplements the prior six months from the study. The dietary characteristics of the patients were not monitored. However, those with malnutritions were not included in the study. Upon your suggestion, we made an addition to the material method section indicating this situation.

                ‘The participants' serum vitamin D levels were evaluated in the autumn. Participants living in the same region had similar levels of sunlight exposure. Sunlight is an important source of vitamin D [19]. The dietary characteristics of the patients were not evaluated. Those with malnutrition were excluded from the study.’

Point 7: 145 line : correct the word SCORAD

correct the Figure 1: the text of the horizontal line 

Response 7: Thank you for your attention. We corrected them.

Point 8: maybe in the discussion section you should also write briefly about the cutaneous immunomodulatory effect of sunlight on skin as vitamin D main source is sunlight  

Response 8: Thank you for your valuable suggestion. Upon your suggestion, these informations are added in discussion section

- According to your recommendation we added ‘Most of the vitamin D is obtained through exposure of the skin to sunlight. Sunlight, especially ultraviolet B (UVB) radiation, has a significant immunomodulatory effect on the skin. Exposure to UVB radiation stimulates the production of vitamin D in the skin, which plays a critical role in immune system regulation [27]. It is observed that at higher geographic latitudes, where there is less sunlight exposure and reduced vitamin D production, there is a higher prevalence of AD [28]. Vitamin D can influence not only calcium regulation but also immunomodulation. Specifically, vitamin D can affect local calcium balance and regulate gene transcription by targeting vitamin D receptors that regulate immune system responses [29]. Additionally, vitamin D has been shown to inhibit the release of markers such as IL-6 and tumor necrosis factor-alpha (TNF-α), which promote inflammation and worsen AD symptoms [30]. Moreover, vitamin D is highlighted for its ability to enhance the production of antimicrobial peptides in the skin, strengthening the skin's natural defense and thereby reducing the risk of infection or exacerbation [31]. Furthermore, it is believed that vitamin D may contribute to the maintenance of skin integrity by supporting the production of critical proteins like filaggrin and involucrin, which are essential for skin barrier function [10].’ in discussion section.

Point 9: also in the discussion section you should add if vitamin D affects eosiniphils count.

Response 9: Thank you for suggestion. We have included a paragraph discussing the effect of vitamin D on eosinophil counts.

According to your recommendation we added ‘Eosinophils are closely linked with the effector component of immune response mediated by type 2 T helper cells, which plays a pivotal role in allergic reactions [43]. Specifically, 1,25(OH) vitamin D has been shown to extend the lifespan of eosinophils and enhance the expression of CXCR4, a receptor involved in chemotaxis and immune regulation [44]. Additionally, it has been observed that vitamin D can reduce eosinophil necrosis and limit the release of cytolytic peroxidase [45]. Nevertheless, the ongoing debate revolves around the association between serum vitamin D levels and the eosinophil count in the bloodstream. While a few studies have established a link between lower vitamin D levels in the blood and elevated eosinophilia, the majority of research has not reported a significant correlation [46-48].’ in discussion section.

Reviewer 2 Report

The study addresses an important issue that may offer valuable information about the association of atopic dermatitis severity with serum vitamin D levels. This question is important because the prevalence of atopic dermatitis is high, especially in children (up to 30%, the authors point out). New ways of relieving symptoms are therefore being sought, and vitamin D supplementation may be one of them. Given the potential of vitamin D to inhibit the inflammatory response, enhance antimicrobial peptide activity and promote the integrity of epidermal barrier permeability, its supplementation offers a possible therapeutic effect in many skin conditions, including atopic dermatitis. However, data on its role in the pathogenesis of allergic diseases are conflicting. Further research into vitamin D supplementation will help to assess whether the 'sunshine vitamin' can be considered as an adjuvant therapy in atopic dermatitis. The authors have therefore revisited this issue.

The research is generally well designed and the article is clearly written.

1.       The main shortcoming of the study seems to be that it did not take into account other factors that can aggravate AD, such as climatic conditions, environmental pollutants or psychosomatic factors. Was a questionnaire administered before the trial to further classify patients? Were patient demographics taken into account?

2.       The discussion should include a paragraph clarifying the potential mechanism of action of vitamin D in AD.

3.       The discussion should also try to better explain the reasons for the discrepancies between the results obtained in this study and between different studies.

4.       Lines 197-198: Please delete the sentence that is repeated.

Author Response

Dear Reviewer,

Thank you for your interest in our manuscript entitled “Evaluation of the Impact of Serum Vitamin D Levels on the SCORAD Index in Pediatric Atopic Dermatitis”.  Our responses are outlined below. Our corrections were also made in the revised manuscript. Corrections were indicated as yellow.

Point 1: The main shortcoming of the study seems to be that it did not take into account other factors that can aggravate AD, such as climatic conditions, environmental pollutants or psychosomatic factors. Was a questionnaire administered before the trial to further classify patients? Were patient demographics taken into account?

Response 1: : Thank you for your valuable suggestions and comments. A questionnaire was not administered to further categorize the patients. The participants' serum vitamin D levels were evaluated in the autumn. The hospital where the study was conducted was a local regional hospital. Participants living in the same region had similar levels of sunlight exposure. We stated that one of the main limitations of our study is that factors that may aggravate atopic dermatitis, such as environmental pollutants or psychosomatic factors, are not taken into account. And we added in Materials and Methods section: ‘The participants' serum vitamin D levels were evaluated in the autumn. Participants living in the same region had similar levels of sunlight exposure. Sunlight is an important source of vitamin D [19]. The dietary characteristics of the patients were not evaluated. Those with malnutrition were excluded from the study.’

Point 2: The discussion should include a paragraph clarifying the potential mechanism of action of vitamin D in AD.

Response 2:

Thank you for your valuable suggestion. Upon your suggestion, these informations are added in discussion section

- According to your recommendation we added ‘Most of the vitamin D is obtained through exposure of the skin to sunlight. Sunlight, especially ultraviolet B (UVB) radiation, has a significant immunomodulatory effect on the skin. Exposure to UVB radiation stimulates the production of vitamin D in the skin, which plays a critical role in immune system regulation [27]. It is observed that at higher geographic latitudes, where there is less sunlight exposure and reduced vitamin D production, there is a higher prevalence of AD [28]. Vitamin D can influence not only calcium regulation but also immunomodulation. Specifically, vitamin D can affect local calcium balance and regulate gene transcription by targeting vitamin D receptors that regulate immune system responses [29]. Additionally, vitamin D has been shown to inhibit the release of markers such as IL-6 and tumor necrosis factor-alpha (TNF-α), which promote inflammation and worsen AD symptoms [30]. Moreover, vitamin D is highlighted for its ability to enhance the production of antimicrobial peptides in the skin, strengthening the skin's natural defense and thereby reducing the risk of infection or exacerbation [31]. Furthermore, it is believed that vitamin D may contribute to the maintenance of skin integrity by supporting the production of critical proteins like filaggrin and involucrin, which are essential for skin barrier function [10].’ in discussion section.

Point 3: The discussion should also try to better explain the reasons for the discrepancies between the results obtained in this study and between different studies.

Response 3: Thank you for your valuable suggestion. Additions have been made to the discussion section and indicated as yellow.

Point 4:  Lines 197-198: Please delete the sentence that is repeated.

Response 4: Thank you for your attention. We fixed it.

Round 2

Reviewer 1 Report

The authors did take my suggestions into consideration The manuscript has been improved significantly and is ready for publication. We'll done!